# Infected connections: Unraveling the impact of a bacterial symbiont on ant-aphid partnership

**Margaux Jossart**[1]*, **Thierry Hance**[2], **Claire Detrain**[1]

**1** Unit of Social Ecology, Université Libre de Bruxelles, Brussels, Belgium, **2** Earth and Life Institute, Biodiversity Research Center, Université catholique de Louvain, Louvain-la-Neuve, Belgium

* Margaux.Jossart@ulb.be

## Abstract

The multitrophic plant-aphid-ant system is a model widely studied in ecology and evolutionary biology. Within this system, bacterial symbionts can circulate and may modify the relationships between partners. A common symbiont of aphids, *Serratia symbiotica,* shows a wide variety of strains with different lifestyles, one of them being associated with the aphid gut and found in the digestive tract of aphid-tending ants. This free-living *S. symbiotica* strain induces fitness costs on its aphid host which can be offset by a lower selective pressure exerted by parasitoids. In this paper, we investigated whether this aphid gut-associated bacterium may alter the mutualistic relationships between *Aphis fabae* aphids and *Lasius niger* ants. Aphids infected with *S. symbiotica* showed a reduced population growth, this negative effect being attenuated in the presence of aphid-tending ants. This bacterium also reduced the ant interest in honeydew-producing aphids: they were less likely to visit plants bearing *S. symbiotica*-infected aphids, they ingested fewer honeydew droplets and they took a longer time before deciding to feed on released honeydew. The bacterium thus makes honeydew less palatable for ant foragers, most probably by altering its composition. This suggests that the free-living *S. symbiotica* strain may promote a gradual abandonment of infected aphids by ants and ultimately jeopardize the ant-aphid mutualistic relationship. We speculate about bacteria-induced consequences of reduced ant protection against aphid natural enemies and increased host plant defense as due to a potential redirection of ant foraging towards extrafloral nectaries as an alternative sugar resource.

## Introduction

The microbiome is known to play a key role in insects, particularly in species that rely exclusively on nutrient-limited foods such as plant sap or blood [1–4]. Aphids feed on plant phloem, which is highly concentrated in carbohydrates but low in amino acids and nitrogen content. Aphids must compensate for this nitrogen shortage by ingesting large amounts of sap, thus rejecting excess sugars in honeydew droplets

**Data availability statement:** All files are available from the Zenodo database. DOI: 10.5281/zenodo.15342394.

**Funding:** M.J. was funded by research project T.0108.21, awarded by the Belgian "Fonds de la recherche scientifique" (FNRS). C.D is research director at FNRS. https://www.frs-fnrs.be/fr/ The funder had no role in study design, data collection and analysis, decision to publish, or preparation of the manuscript.

**Competing interests:** The authors have declared that no competing interests exist.

[5,6], and by harboring endosymbiotic bacteria that synthesize essential amino acids [7,8]. Almost all aphid species host *Buchnera aphidicola* bacteria, which are obligate symbionts as they provide essential nutrients lacking in the insect diet [1,7,8]. Due to the long-term association with the aphid host, these symbionts have genomes that can be drastically reduced, thereby severely hampering their capability to survive outside of their host [9]. Obligate symbionts are hosted in specialized cells called bacteriocytes and are transmitted strictly vertically from mother to offspring [9–11]. Aphids may also harbor facultative endosymbionts that are not essential for their survival and reproduction. These facultative endosymbionts can inhabit sheath cells, hemolymph, secondary bacteriocytes, or the digestive tract [12–14]. Facultative symbionts are usually transmitted vertically, although occasional horizontal transfers may occur among aphids [14,15] or may be acquired from environmental sources [14,16,17]. Harboring facultative symbiotic bacteria can confer to insect hosts new abilities to conquer ecological niches [18]. Indeed, several studies have highlighted the importance of facultative bacteria in the acquisition of ecologically and physiologically important traits by the aphids [19–25], such as improved heat tolerance [21], enhanced protection against a range of natural enemies [14,19,22,26,27] increased diet breadth [2,23,24], or improved survival on less favorable plants [25]. Although facultative symbionts can have beneficial effects on the aphids, they can also impose fitness costs by reducing aphid's longevity and reproductive success [14,28–30], and can affect host physiology by reducing the body mass of aphids [14]. Bacteria can also incur indirect costs by altering the behavioral defenses of their aphid host, such as a lower propensity to drop off the plant when being stressed [31] or a reduction in defensive behavior (e.g., rapid movement of their legs) towards predators and parasitoids [32]. As for the symbiotic bacteria, living in aphid hosts provides them with access to food, favorable conditions for population growth, and a means of dispersal through both inheritance and horizontal transfer [19,33]. The microbiome associated with aphids can also influence the ecology of aphids and their relationships with other insects [19]. Bacteria such as *Staphyloccocus spp.* found in aphid honeydew have an impact on higher trophic levels. They can be beneficial to the aphids by making them more attractive to tending ants [34], but they can also be detrimental to hemipterans by attracting their natural enemies [35]. This example illustrates the complexity and sometimes the opposite effects that bacteria may have in multitrophic systems.

The mutualism between ants and aphids has been extensively studied, namely due to the crop pest status of several aphid species and the key role played by ants in agricultural and natural ecosystems [34,36–40]. The ant-aphid interaction is considered a key factor in shaping aphids' evolution, particularly in aphid species that are obligatorily dependent on ants [40,41]. This long-lasting relationship has influenced co-evolutionary pathways, resulting in morpho-functional and/or behavioral adaptations in both aphids and ants [38–41]. Ant-aphid mutualism is based on reciprocal services: sap-feeding hemipterans provide honeydew, while ants clean the aphid colony of excess honeydew and protect them from predators and parasitoids [38,39,42]. As a result, ant-tended aphid populations typically exhibit increased developmental rates and faster growth than unattended populations [43–45]. However, it should be

noted that ants can occasionally switch to predation when aphid population becomes too dense [38,46]. Mutualistic relationships are initiated by ant foragers, which may discover aphids by chance or which may be attracted from a distance by volatile organic compounds (VOCs) present in aphid honeydew [34], or by low amounts of aphid alarm pheromone [47]. Once the aphid colony has been discovered, the level of ant attendance is positively correlated with the amount of released honeydew and with its sugar composition. In particular, honeydew content in polysaccharides such as melezitose and sucrose, is known to trigger the laying of a recruitment trail by ant foragers, thereby leading to a collective exploitation of aphids [37,38,40,48]. Although myrmecophilous aphids provide a local and relatively stable source of energy for the ants [38,48], ant-aphid mutualism often remains a facultative relationship of which the maintenance strongly depends on a positive cost-benefit balance for both partners [38–41]. Any significant reduction in the net benefit received by one species, or the advent of a new constraint that disrupts ant-aphid interactions, makes facultative mutualists susceptible to partner switching or a shift towards antagonistic relationships. Specifically, the stability of mutualistic interactions with ants could be reinforced or threatened if changes in honeydew or aphid behavior are induced by the microorganisms associated with hemipterans.

The bacterium, *Serratia symbiotica*, is an endosymbiont commonly found in natural populations of aphids [22,49,50]. The different *S. symbiotica* strains exhibit variable degrees of symbiosis with their aphid hosts. The co-obligatory strains, associated within the aphid subfamily Lachninae, are located in bacteriocytes [51] and have a nutritional role, compensating for primary symbionts which can no longer produce the amino acids required by their aphid host [51,52]. As for the facultative strains of *S. symbiotica*, they are not essential for aphid survival but have nevertheless beneficial effects such as conferring protection against parasitoids or improving heat tolerance of aphids from the subfamily Aphidinae [21,53]. They can be intracellular [12,22,54] or extracellular with a free lifestyle and a weak dependence on their aphid host [14]. Some of these free-living facultative strains are gut-associated [13,14,55]. In the case of *S. symbiotica* bacteria, it has been suggested that those gut-associated strains are in the early stages of symbiotic life with aphids and could be ancestors of several facultative and co-obligate strains [14,51]. In addition, a gut-associated *S. symbiotica* strain (CWBI-2.3$^T$) has been isolated from the black bean aphid, *Aphis fabae* [56], and cultured on an artificial medium [56–59], allowing manipulation of its occurrence in each species of the host plant-aphid-ant system. This aphid gut-associated bacterial strain is transmitted horizontally. It can propagate through the population of sap-sucking insects by feeding on *S. symbiotica*-infected phloem of the host plant [60] or by contamination of aphid honeydew [14]. This strain was even found in the gut of ants tending aphids [13,50]. This free-living strain is known to have a fitness cost for *A. fabae* aphids by reducing their fecundity and lifespan, but also to be a defensive symbiont by providing increased protection against parasitoids [14].

In addition to investigating the influence of bacterial infection on the direct fitness of the aphids, one may ask whether these facultative symbionts make *S. symbiotica*-infected individuals more or less attractive to the ant partners. If the gut-associated *S. symbiotica* strain reinforces the attractiveness of aphids for the ants, the bacterium would confer a fitness gain to infected aphids by inducing a higher level of attendance by the ants. In this case, we speculate that infection by the free-living strain of *S. symbiotica* may have promoted, on an evolutionary timescale, a shift towards a more obligate form of symbiosis, leading to a more intimate relationship between the bacterium and its aphid host. The alternative hypothesis of a bacterium that reduces the attractiveness of aphids to ants, suggests that this symbiont may facilitate a disruption of the mutualistic ant-aphid interaction. In this latter scenario, the facultative *S. symbiotica* strain would exert a double negative effect on aphids: a direct cost of reduced fitness [14,59] and an ecological cost on their partnership with ants. For its part, the bacterial strain may eventually suffer from reduced dispersal and abundance within aphid populations as a result of the net costs it imposes on its aphid host.

In this study, we investigated the impact of the free-living facultative bacteria, *S. symbiotica,* on the growth of *A. fabae* colonies as well as its effects when aphids were tended by ant mutualists. To this aim, we compared the growth dynamics of *S. symbiotica*-free and *S. symbiotica* (CWBI-2.3$^T$)-infected aphid populations, being tended or not by *Lasius niger* ants. Furthermore, we quantified to which extent the ants foraged and fed on the honeydew of *S. symbiotica*-infected aphids,

kept interest in these aphids in the long run and provided them with care, thereby influencing the strength of their mutualistic relationships.

## Materials and methods

### Insect rearing and bacterial strains

The common black garden ant *Lasius niger* is a well-known aphid-tending species that is widespread in temperate regions of Europe. Colonies were harvested in the south of Brussels (50°49'05.1"N, 4°24'01.6"E) and kept in the laboratory under controlled conditions: 21±1°C and 46±5% relative humidity, with a 12:12 light: dark photoperiod. Ants were reared in 43 x 29 x 8 cm plastic containers whose edges were covered with polytetrafluoroethylene (Fluon™, Withford, U.K.) to prevent ants from escaping. Tubes filled with water and an aqueous sucrose solution (0.3M) as well as dead house crickets (*Acheta domesticus*) were provided as protein source and replaced every week. The nest consisted of test tubes, of which one half was filled with water and plugged with cotton wool to provide a humid atmosphere for the ants. The test tubes were covered with a red transparent paper to create darkness inside the nest. *A. fabae* aphids (clone A06-407 provided by Dr. C. Vorburger, ETH Zürich, Switzerland) were used in this study. This clone was diagnosed as being uninfected with facultative endosymbionts by diagnostic PCR [29,61], thereby providing an ideal biological model to isolate the impact of a facultative bacterium of interest following an experimentally-induced infection. Aphid clones were raised on *Vicia faba* plants in insect cages (BugDorm-4M4545D) at 20±1°C under horticultural lighting (BL-SPECTRAL-27–120+) with a photoperiod of 16h light and 8h dark. *V. faba* plants were replaced regularly to maintain the growth of aphid populations. The plantlets were grown in potting soil (Potting Soil CDM 1004522) with the same photoperiod and temperature as aphids in insect cages and were watered twice a week.

The CWBI-2.3$^T$ cultivable strain of *S. symbiotica* isolated from *A. fabae* [55,56] was used for the experiments. This strain is a free-living bacterium associated with the gut of aphids and, as it circulates throughout the tripartite ant-aphid-plant system, it can affect the fitness of each partner. The bacterium was preserved in a frozen stock at −80°C and cultured at 28°C in 868 Agar (1% yeast extract, 1% casein peptone, 1% glucose, 1,7% agar) as described by Sabri et al. [56]. Bacteria colonies were sub-cultured every 3 weeks.

### Oral infection of aphids

Before carrying out the experiments, several groups of *A. fabae* aphids were infected by a *S. symbiotica* solution by using the following procedure. First, in 50 ml centrifuge tube, the bacterium was let to grow to an early log phase on 863 medium (1% yeast extract, 1% casein peptone, 1% glucose) [56] on a gyratory shaker (160 rpm) at 20°C, until reaching an optical density (OD) between 0.5 and 0.7 at 600nm during the logarithmic growth phase. Afterward, bacteria were centrifuged (4000 rpm) then washed with sterile phosphate-buffered saline (PBS; Sigma). Finally, they were suspended in PBS to obtain a bacterial suspension with an OD of 1 at 600nm. Fifteen microliters of this bacterial culture were added to 1.5 ml of artificial diet. The artificial diet used was a sugar water solution with amino acids, it was based on protocols found in the literature [62–65]. This mix corresponded approximately to a bacterial concentration of $10^6$ cfu/ml [14,59]. Oral infection was performed by feeding aphids for 24h in a setup aimed to imitate a leaf: two parafilm squares were stretched on a Petri dish (33 mm) between which we placed 200 µL of *S. symbiotica*-infected diet. *S. symbiotica*-infected aphids were then transferred to *V. faba* plants (physiological stage BBCH12). These plants were watered with a bacterial suspension of $10^9$ CFU/ml, for 5 days to enhance the colonization of the digestive tract of aphids and to ensure a bacterial contamination of honeydew [14]. After 5 days, aphids were placed on experimental plants of similar age and physiological stage (*V. faba*, 15 days, BBCH12), watered only with water. The same procedure was applied to control groups of *S. symbiotica*-free aphids, except that sterile PBS (instead of bacterial suspension) was added to the artificial diet.

Diagnostic PCR was conducted to confirm the presence of cultivable *S. symbiotica* in aphids following their oral infection, and to check the uninfected status of the *S. symbiotica*-free aphids. These aphids were collected from the patches

subsequently used in the experiments. PCR analyses were carried out on 10 aphid individuals, with five collected from the uninfected patch and five from the *S. symbiotica*-infected patch. All 10 individuals were sampled on the 5th day following oral infection, coinciding with the transfer of infected and uninfected aphids to experimental plants. Although *S. symbiotica* bacteria is known to persist and even proliferate over time within *A. fabae* digestive track [14], the infection status of aphids was reassessed 20 days post-infection by the artificial diet in order to confirm its stability. At this point, diagnostic PCR was performed on 10 aphid individuals per experimental condition (*S. symbiotica*-infected aphids with or without ant attendance as well as uninfected aphids with or without ant attendance).

DNA from individual aphids was extracted and purified by using Wizard® Genomic DNA Purification kit (Promega). PCR primers used for *S. symbiotica* detection were 16SA1 (5'-AGAGTTTGATCMTGGCTCAG-3') and PASScmp (5'-GCAAT GTCTTATTAACACAT-3') as described by Fukatsu et al. [66]. PCR reactions were performed in a 25 µl volume containing 1 µl of DNA solution, 1.25 µl of 10 µM of each primer, 12.5 µl of 2x Taq Master Mix (Apex Bioresearch Products) and ultra-pure water. Thermal cycling conditions included 35 cycles at 95°C for 30 s, 55°C for 40 s and 72°C for 30 s. To confirm the bacterial strain used for infection, PCR were also performed on frozen stocks of CWBI-2.3T cultivable strain, as well as on samples of the infected artificial diet used for the oral infection of aphids. The resulting PCR products were sequenced by Macrogen (Netherlands).

### Impact of bacteria on the growth dynamics of aphid populations

To observe the effects of the CWBI-2.3T strain on aphid fitness, we conducted a 15-days experiment comparing the growth dynamics of aphid populations infected or not *with S. symbiotica*. Both populations were left unattended by ants (Unattended *S. symbiotica*-infected aphids: USInf and Unattended *S. symbiotica*-free aphids: USFree). We also investigated whether ant attendance had a different impact on aphids depending on their infection status. Thus, we monitored the growth of *S. symbiotica*-infected and uninfected aphid populations in the presence of their ant partners (Attended *S. symbiotica*-infected aphids: ASInf and Attended *S. symbiotica*-free aphids: ASFree). Ten aphid colonies were monitored for each treatment (USInf, USFree, ASInf, ASFree). For the *S. symbiotica*-infected populations, 60 adult aphids (wingless) were orally infected by the bacterium (as described above). Five days post-infection, 30 aphids were placed on one plant in contact with ants (ASInf). The other 30 aphids were placed on another plant not exposed to ants (USInf). The same procedure was followed for the Uninfected populations, where 60 aphids were fed on an artificial diet with sterile PBS and were divided between two plants, one being visited by the ants (ASFree) and the other being free of any ant worker (USFree). Every day, the number of adult aphids and late nymphal instars (3rd and 4th instars) was counted on each of the four plants. Although *A. fabae* undergoes four nymphal stages, aphid counts started from the third nymphal instar because early instars (1st and 2nd) were extremely small, morphologically similar and difficult to count accurately by eye. In addition, the experimental setup could not be dismantled to remove, identify and count all aphids, including early instars, without causing significant disturbance to the plants and experimental procedure. To maintain consistency, the same observer performed all daily counts from the third instar onward. The size of the stem (from base to apex) and the phenological state of the plant based on Lancashire et al. [67] were censused during the entire experiment.

### Impact of *S. symbiotica* on interactions between ants and aphids

For the conditions where aphids were tended by the ants, we used ten experimental queenless and broodless ant colonies, each consisting of 500 *L. niger* workers (400 internal workers and 100 foragers). Before carrying out the experiments, ant colonies were placed in a 54 x 38 x 8 cm plastic container and were food deprived for four days. Then, thirty minutes before the beginning of the experiment, two plants (BBCH12) were placed in the foraging area of an ant colony, 15 cm apart and 40 cm from the nest tubes occupied by the ant workers. One plant was infested by 30 *S. symbiotica*-infected aphids and the other one hosted 30 *S. symbiotica*-free aphids. Each plant pot was placed in a Petri dish of which the sides were covered with Fluon™ and was sealed with parafilm. This prevented ants from digging into the soil

or climbing on the plant before the start of the experiment. Each trial started when a 30 cm Y-shaped bridge was added in the setup to connect the ant colony to the aphid-infested plants and lasted for 10 days (S1 Fig). The experiments were carried out in transparent polycarbonate climatic chambers (120x70x60 cm) under controlled conditions of temperature and hygrometry (20 ± 1°C and 55% relative humidity).

## Collective foraging of ants on aphids

Ants were allowed to simultaneously exploit a colony of *S. symbiotica*-free aphids and *S. symbiotica*-infected aphids. All the measures were taken three times a day (at 10am, 1 pm and 4 pm). The flows of ants towards each plant were recorded, with a camera (Logitech HD Pro C920), on each branch of the Y-shaped bridge, for 15 minutes. An ant was counted as soon as it crossed an arbitrary line, which was located on both branches of the bridge at 2 cm from the plant stem, by twice the length of its body. In addition to these flows, we counted the number of ants in the foraging area, on the bridge and on each plant (stem, leaves and pot) as well as the number of workers contacting aphids. Out of these data, we calculated the total number of foragers, a plant occupancy index (ratio of ants located on the plants over the total number of foragers outside their nest), an exploitation index of the plant hosting *S. symbiotica*-infected aphids (ratio of ants located on the plant bearing *S. symbiotica*-infected aphids over the total number of ants on both plants) and finally, an interaction index between ants and *S. symbiotica*-infected aphids (ratio of the number of ants contacting *S. symbiotica*-infected aphids over the total number of ants in contact with aphids on both plants). For all counts, we averaged the three measurements taken per day to obtain daily values.

## Aphid tending behavior and honeydew consumption by ants

At a local scale, we compared how the ants interacted with *S. symbiotica*-free or *S. symbiotica*-infected aphids. On each plant, an aggregate of 5 aphids on average (group of minimum 4 and up to 10 adult individuals) was filmed using a fixed camera (4K Camera ELP) three times a day (at 10am, 1 pm and 4 pm) for 15 minutes. The recorded images covered an area of around 25 cm² of leaf surface. These recordings were made until the 5th day of the experiment, which cor-responded to 10 days after the initial infection of aphids by *S. symbiotica*. Behavioral data were collected by using the software BORIS [68] and were used to quantify the number of contacts made by ants to stimulate the release of honey-dew droplets, the production of honeydew droplets as well as the feeding response of tending ants. We also paid atten-tion to any agonistic interactions of ants towards aphids, including predation, bites or threatening behavior. We estimated the honeydew production by adding the number of droplets that were spontaneously emitted and those released after an antennal stimulation by ants. We also counted the number of honeydew droplets that were collected by the ants, that were not collected and withdrawn back in the aphid digestive tract or that dropped over the ground. For all these counts of honeydew droplets, we averaged the three measurements taken per day to obtain daily values. To assess the propensity of aphids to release honeydew and the eagerness of ants to feed on, we measured the time elapsed between the release of a honeydew droplet and its collection by an ant. We also measured the duration of the ant's antennation before the stimulated aphid released a honeydew droplet.

## Statistical analyses

All data were analyzed with R software (R 4.3.0) and all results were considered as significant when the associated p-value was lower than alpha = 0.05. The normality of the data was assessed using both statistical and graphical meth-ods. First, the Shapiro-Wilk test was conducted to evaluate the normality of the residuals. Additionally, we examined quantile-quantile (Q-Q) plots to visually inspect the distribution of the residuals against a theoretical normal distribution. These combined approaches provided a comprehensive evaluation of the assumption of normality for our data. When multiple pairwise tests were performed, a Bonferroni correction was used to adjust the p-values. Data were expressed as mean ± standard deviation and all figures were plotted using the "*ggplot2*" package [69].

We carried out LMMs, GLMs and GLMMs with the R-package "*lme4*" to analyze data that met the model's assumption and that showed no under or over-dispersion based on model deviance/degrees of freedom values [70,71]. Overdispersion was tested for all analyses. For all analyses, the most comprehensive model was considered, including all fixed factors and possible interactions. To select the best fitted model based on the minimum Akaike Information Criterion (AIC), we used the R-package "*MASS*" with the "*step*" function and the R-package "*MuMIn*" [72] with the "*dredge*" function to display the performance indices of the models. We used the R-package "car" for the significance of fixed factors and possible interactions [73]. When significant, we conducted post-hoc comparisons by using Tukey tests with the R-package "*emmeans*" [74].

## Growth dynamics of aphid populations

Data from the 4 different aphid populations (USFree, USInf, ASFree, ASInf) were not distributed normally on certain days (Shapiro-Wilk tests). The growth dynamics of aphid populations were thus analyzed using generalized linear models (GLM) with a negative binomial error structure and a log-link function. We included the infection status of aphids, time and the presence of ants as fixed factors and the number of aphids as dependent variable (S1 Table). The best model considered only the 3 fixed factors (S2 Table). The significance of factors was tested using Wald Chi Squared tests. We conducted Wilcoxon signed-rank tests to evaluate the impact of bacterial infection on aphid numbers, by comparing uninfected and *S. symbiotica*-infected aphid populations each day for both unattended (USFree vs. USInf) and ant-attended (ASFree vs. ASInf) populations. Likewise, we examined the effect of ant attendance by comparing each day, using Wilcoxon signed-rank tests, the sizes of uninfected (USFree vs. ASFree) aphid populations as well as the sizes of *S. symbiotica* infected (USInf vs. ASInf) aphid populations. The p-values were adjusted using the Bonferroni correction to account for multiple comparisons.

## Collective foraging of ants on aphids

The impact of time and bacterial infection of aphids on the ant's ascending flows to the plant was tested using a generalized linear mixed model (GLMM) with a negative binomial distribution and a log-link function. Ant colonies were used as random factor. The time and *S. symbiotica*-infection status of aphids were considered as fixed factors (S1 Table). The best model considered the two fixed factors (S3 Table). The significance of factors was tested using Wald Chi Squared tests. As the numbers of ants counted over the plant, on the foraging area or in contact with the aphids were distributed normally, we performed one-factor repeated measures ANOVAs to check if time influenced the number of foragers, with the ant colonies treated as the within-subject factor. Sphericity was tested using Mauchly's test and, when required, a Greenhouse-Geisser correction was applied. By using a Student t-test, we also compared the exploitation index and the interaction index of ants with *S. symbiotica*-infected aphids to a theoretical value of 0.5 expected if the ants exploited equally both aphid colonies regardless of their infection by bacteria.

## Aphid-tending behavior and honeydew consumption by ants

Since the number of honeydew droplets released by aphids were normally distributed, two-way repeated measures ANOVAs were used to test the effect of time and infection status of aphids on the number of honeydew droplets, the ant colonies being included as within-subject factor. The ANOVAs were performed to test the main effects of each factor as well as interactions. Mauchly's test was used to assess the assumption of sphericity, and, when violated, the Greenhouse-Geisser correction was applied to adjust the degrees of freedom. When significant, pairwise comparisons of released honeydew droplets between levels of each factor were performed using Tukey post hoc tests.

Concerning the ant feeding responses to honeydew droplets emitted by aphids, we calculated a consumption index. This index is the proportion of honeydew droplets consumed by ants among those that were emitted by aphids following a stimulation by ants. The consumption index was analyzed under a generalized linear model (GLMM) with a beta binomial

error structure and a log-link function. The time spent by ants stimulating an aphid with its antennae was analyzed by a general linear mixed model (LMM), both the symmetry and normality of residuals were assessed to ensure the validity of the model. For those two models, the fixed factors were the time and infection status of aphids, while the ant colonies were treated as the random factor (S1 Table). Each of the two best models included the two fixed factors as well their first-order interaction effect (S4 and S5 Tables). The significance of fixed factors was tested using Wald Chi Squared tests. After fitting the model, we examined pairwise contrasts between levels of the day and infection status, with p-values adjusted using the Tukey method for multiple comparisons. In order to estimate the eagerness of foragers to feed on honeydew, we calculated the proportion of honeydew droplets not yet consumed as a function of the time elapsed since their release by the aphids. These survival curves were compared for *S. symbiotica*-free and *S. symbiotica*-infected aphids using a log-rank test (with R-package "survival").

## Ethical note

This study complies with the local legislation regulating scientific research on animals. No license or permit was required for this research. Care was taken to minimize adverse effects on the welfare of tested colonies. Ant colonies were provided with suitable nesting sites, food and water. The experiments were carried out with no obvious harm to the ants and without any significant increase in the mortality of workers. After the experiments, the ant colonies were maintained in the laboratory and reared until their natural death.

## Results

### Impact of bacteria on the growth dynamics of aphid populations

The size of aphid populations remained stable for the first 3 days, including the 30 individuals that were initially placed on the plant. Thereafter, the average number of aphids rapidly increased until it reached maximum values on the last two days of the experiment (on day 14 for three experimental populations: ASFree: $199,6 \pm 49,7$ aphids; ASInf: $166,0 \pm 40,8$ aphids and USInf: $131,4 \pm 23,6$ aphids and on day 15 for USFree: $160,9 \pm 36,6$ aphids; Fig 1). There was a significant time effect, with all aphid populations growing over the 15 days (GLM: time effect: Wald $\chi^2$ test $= 3136.4$, df $= 14$, p $< 0.001$). For all aphid populations, there was no significant change in the number of aphids for the first five days, then the number of aphids increased significantly up to day 9 (Tukey post-hoc tests, p $< 0.05$; S6 Table). The absence of counts of 1st and 2nd nymphal instars likely explains why significant population growth was only observed starting on day 5, when the earliest nymphs reached the later developmental stages. Aphid populations remained quite stable from day 10 until day 15. We also found a significant effect of bacterial infection since the aphid populations were smaller and grew at a slower rate when they hosted *S. symbiotica* (GLM: *S. symbiotica*-infection effect: Wald $\chi^2$ test $= 87.1$, df $= 1$, p $< 0,001$). The population size of uninfected aphids was significantly higher than that of *S. symbiotica*-infected aphids from day 4 for the ant-attended populations (ASFree vs ASInf) until the end of the experiment (Wilcoxon tests, p $< 0.05$, S6 Table) and from day 4 for the unattended aphid colonies (USFree vs USInf) until day 12 of the experiment (Wilcoxon tests, p $< 0.05$, S6 Table). As for the presence of ants, it had a significant and positive effect on the growth of aphid populations regardless of their infection status (GLM: ants' presence effect: Wald $\chi^2$ test $= 32.0$, df $= 1$, p $< 0.001$). When considering uninfected aphid populations, attendance by ants had a time-limited beneficial effect, with ASFree aphids being significantly more numerous than USFree ones only on day 12 (Wilcoxon tests, p $< 0.05$, S6 Table). The positive impact of ant attendance on the growth of aphid colonies was more pronounced when they were infected by *S. symbiotica,* with ASInf aphids being more numerous than USInf aphids on days 7, 8, 11 and 13 (Wilcoxon tests, p $< 0.05$, S6 Table). The beneficial effect resulting from ant attendance somewhat compensated for the genuine lower growth of *S. symbiotica*-infected aphids. Indeed, populations of infected aphids accompanied by ants (ASInf) showed a growth dynamics close to that of uninfected and unattended aphid populations (USFree), and even caught up with them in the last 3 days (Fig 1).

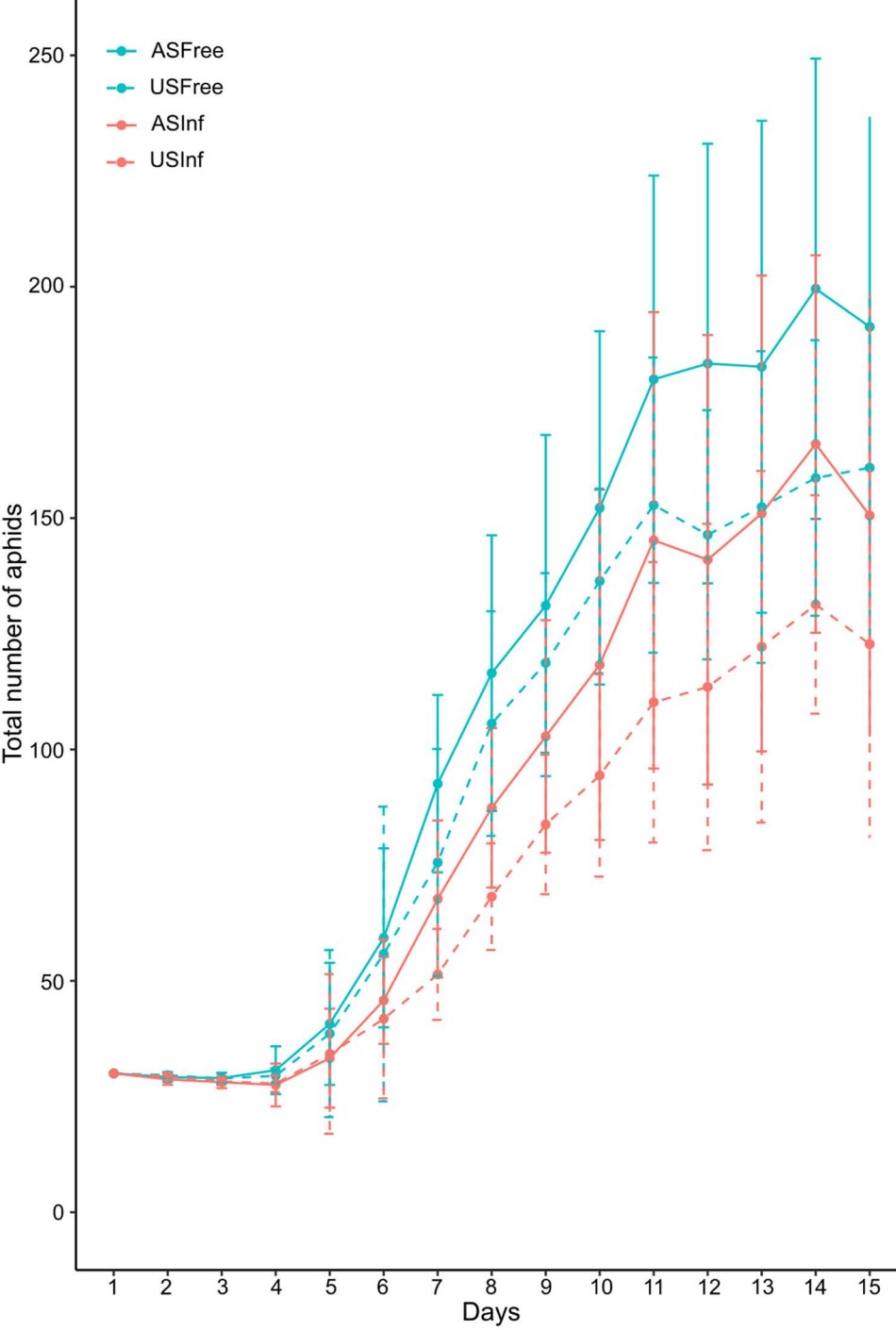

**Fig 1. Growth dynamics of *A. fabae* colonies depending on their *S. symbiotica* infection status and the presence of tending *L. niger* ants.** The growth of aphid populations was censused daily over 15 successive days. Curves are drawn for the different experimental conditions. Blue curves: uninfected aphid populations, red curves: *S. symbiotica*-infected aphid populations, solid curves: ant-attended aphid populations, dashed curves: unattended aphid populations (ASFree: Attended *S. symbiotica*-free aphids; USFree: Unattended *S. symbiotica*-free aphids; ASInf: Attended *S. symbiotica*-infected aphids; USInf: Unattended *S. symbiotica*-infected aphids). Average values are represented by points and were calculated over 10 aphid colonies for each population, error bars represent standard deviation.

## Collective foraging by ants

Just before the start of the experiment, an average of 84,0 ± 16,1 ants walked over the foraging area (n = 10 ant colonies). As soon as the plants were connected by the Y-shaped bridge to the foraging area of ant colonies, foragers climbed over it, and within less than 3 minutes, started to explore both plants simultaneously. The daily ascending flows of ants were slightly lower towards the plant hosting *S. symbiotica*-infected aphids but were not statistically different from the flows towards the plant bearing uninfected aphids (GLMM: *S. symbiotica*-infection effect: Wald χ² test = 0.9, df = 1, p = 0.3). In both cases, the highest flows of foragers over 15 minutes were observed on the first day of the experiment, with an average of 23.3 ± 15.5 (SD, n = 10) and 23.9 ± 14.2 (SD, n = 10) ants walking towards the ASFree and ASInf aphids respectively (S2 Fig). Then, over time, the ascending ant flows significantly decreased regardless of the infection status of aphid colonies (GLMM: time effect: Wald χ² test = 87.6, df = 9, p < 0.001).

The plant occupancy index also changed over time (repeated measures ANOVA I: F (3,31)=6.3, p = 0.001). Among all the ants being outside the nest, the percentage of foragers located over the plants hosting aphid colonies approximated 30% and remained stable until day 3 (Fig 2A, green curve). Then, ant interest in exploiting sap-sucking insects progressively decreased over time until day 10, down to only 16% of foragers. This is likely resulting from a reduced demand for food in ant colonies due to the absence of brood. The exploitation indices of *S. symbiotica*-infected aphids did not change significantly over the course of the experiment (repeated measures ANOVA I: F(9,72)=1.62, p = 0.1). Interestingly, these exploitation indices were always lower than the theoretical value of 50% expected from an equal exploitation of both aphid colonies (Fig 2A, red curve). This under-exploitation of *S. symbiotica*-infected aphids was the most pronounced on days 3,6,7 and 8 (indices significantly lower than 50%; two-tailed t-test, p < 0.05; Fig 2A). These low exploitation indices resulted from lower, although not significantly different, foraging flows to the plant with *S. symbiotica*-infected aphids and possibly from foragers staying for a shorter time on this plant. As for the interaction indices, they did not change over time (repeated measures ANOVA I: F(2,15)=2.5, p = 0.1; Fig 2B), and were also below the theoretical value of 50% for the first 5 days of experiments. The relative number of foragers in contact with *S. symbiotica*-infected aphids was significantly lower than expected from an equal interest in ASFree and ASInf aphids, on day 3 and 4 (indices significantly lower than 50%; two-tailed t-tests, p < 0.05; Fig 2B). This confirms that ants were less eager to stay on plants and interact with aphid colonies infected by *S. symbiotica* bacteria (Fig 2).

## Aphid-tending behavior and honeydew consumption by ants

We examined whether infection by *S. symbiotica* bacteria influenced honeydew production by sap-feeding insects. There was no significant main effect of the infection status (repeated measures ANOVA II: *S. symbiotica*-infection effect, F (1,9) =3, p = 0.1) meaning that an infection by *S. symbiotica* did not alter the total honeydew production by *A. fabae* aphids. However, the number of released droplets increased significantly over time for both infected and uninfected aphids (repeated measures ANOVA II: time effect, F(2,15)=11.4, p = 0.001; Fig 3A), with a significant difference between the first 2 days and the last 3 days of the experiment (Tukey post-hoc tests, p < 0.05; Fig 3A). There was no significant interaction between infection status of aphids and time for the total number of droplets released by the aphids (repeated measures ANOVA II: interaction effect, F(2,18)=0.2, p = 0.8). Considering only the honeydew droplets that were spontaneously emitted per aphid, their number significantly increased over the 5 first days (repeated measures ANOVA II: time effect, F(4,36)=19.4, p < 0.001; Table 1) and were more frequently released by *S. symbiotica*-free than *S. symbiotica*-infected aphids (repeated measures ANOVA II: *S. symbiotica*-infection effect, F(1,9)=8.6, p = 0.02; Table 1).

The number of honeydew droplets released per aphid following an active stimulation by ants did not significantly differ depending on aphid infection status (repeated measure ANOVA II: *S. symbiotica*-infection effect: F(1,9)=0.1, p = 0.7; Table 1) and it did not change over time (repeated measure ANOVA II: time effect: F(4,36)=0.3, p = 0.7; Table 1). The interaction effect between time and infection status on this number of released honeydew droplets was also

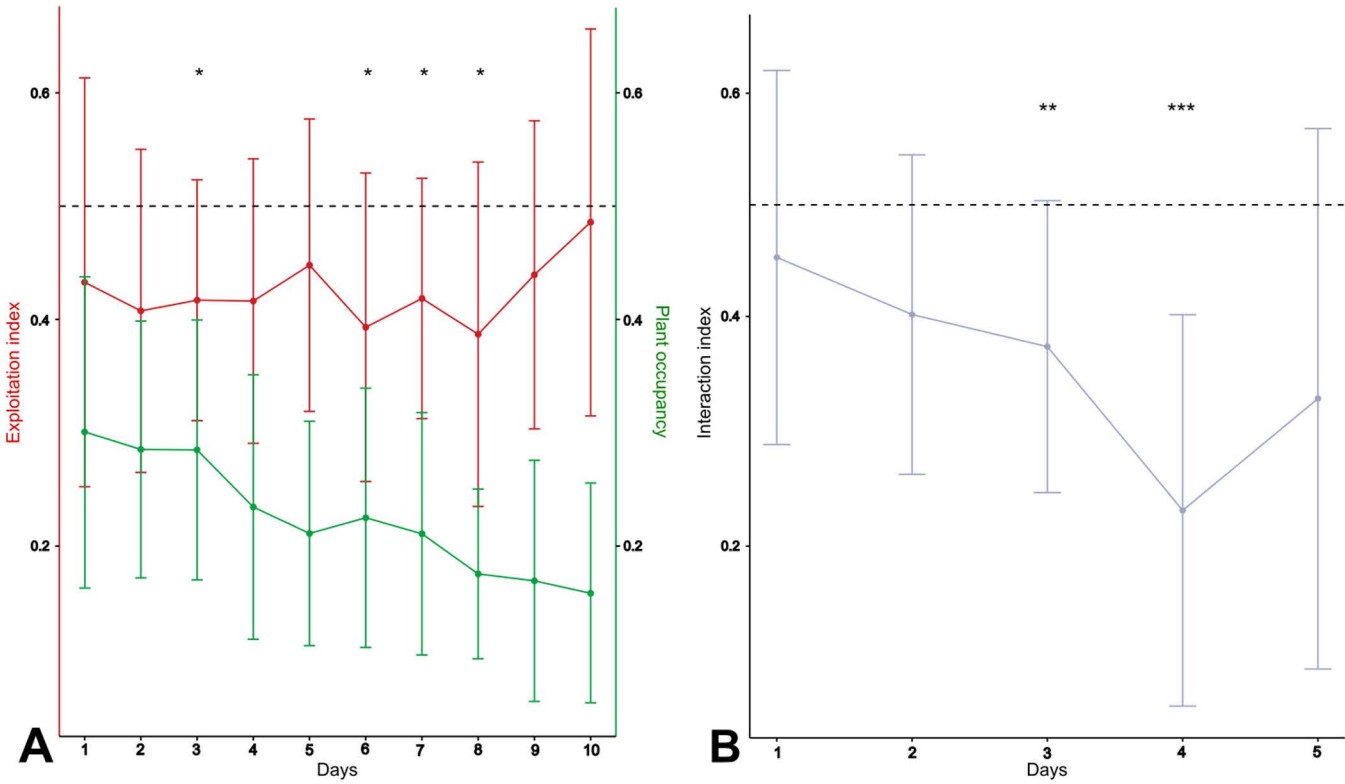

**Fig 2. Collective exploitation of aphid colonies by ants over the course of the experiment.** (A) Exploitation index of *S. symbiotica*-infected aphids by the ants (red curve, left y axis) and plant occupancy index (green curve, right y axis), values were averaged over the 10 *L. niger* colonies. (B) Interaction index of ants with colonies of *S. symbiotica*-infected aphids. Values were averaged over only 9 ant colonies (no contact with infected aphids occurred for one colony). This index was calculated only for the 5 first days (D1-D5) due to the too few contacts with infected aphids for the 5 last days (D6-D10). Error bars represent standard deviation. The black dotted lines stand for the value of 0.5 corresponding to an equal exploitation (2A) or an equal level of interaction with aphids (2B) regardless of their infection by *S. symbiotica*. Indices significantly lower than expected value of 50% (*, p < 0.05; **, p < 0.01; ***, p < 0.001).

not significant (repeated measure ANOVA II: interaction effect: F (4,36) =0.3, p = 0.8; Table 1). Furthermore, ant foragers spent a similar amount of time performing antennal contacts with uninfected and infected aphids (LMM: *S. symbiotica*-infection effect: Wald $\chi^2$ test = 3.5, df = 1, p = 0.06; S7 Table). We however found a significant time effect (LMM: time effect: Wald $\chi^2$ test = 11.5, df = 4, p = 0.02; S7 Table) and interaction effect (LMM: interaction effect: Wald $\chi^2$ test = 53.2, df = 4, p < 0.001; S3 Fig and S7 Table) on the duration of antennal contacts. Indeed, ants made antennal contacts with uninfected aphids for an increasing amount of time over the 5 first days while there was a tendency for a decreasing duration of contacts with *S. symbiotica*-infected aphids. Antennal contacts with aphids were the only interactions made by ants, which never displayed any agonistic behaviors or predation towards *S. symbiotica*-infected or uninfected aphids.

While the production of honeydew droplets was similar for infected and uninfected aphids, the ants did not consume them equally. Indeed, the proportion of droplets that were ingested by foragers was significantly higher when they had stimulated *S. symbiotica*-free aphids compared to bacteria-infected ones (GLMM: *S. symbiotica*-infection effect: Wald $\chi^2$ test = 11.8, df = 1, p < 0.001; S7 Table). There also was a significant effect of the time (GLMM: time effect: Wald $\chi^2$ test = 11.2, df = 4, p = 0.02; S7 Table), with a decrease in the proportion of droplets consumed by ants after having stimu-lated their aphid partner (Fig 3B). There was a significant interaction between infection status and time for the proportion

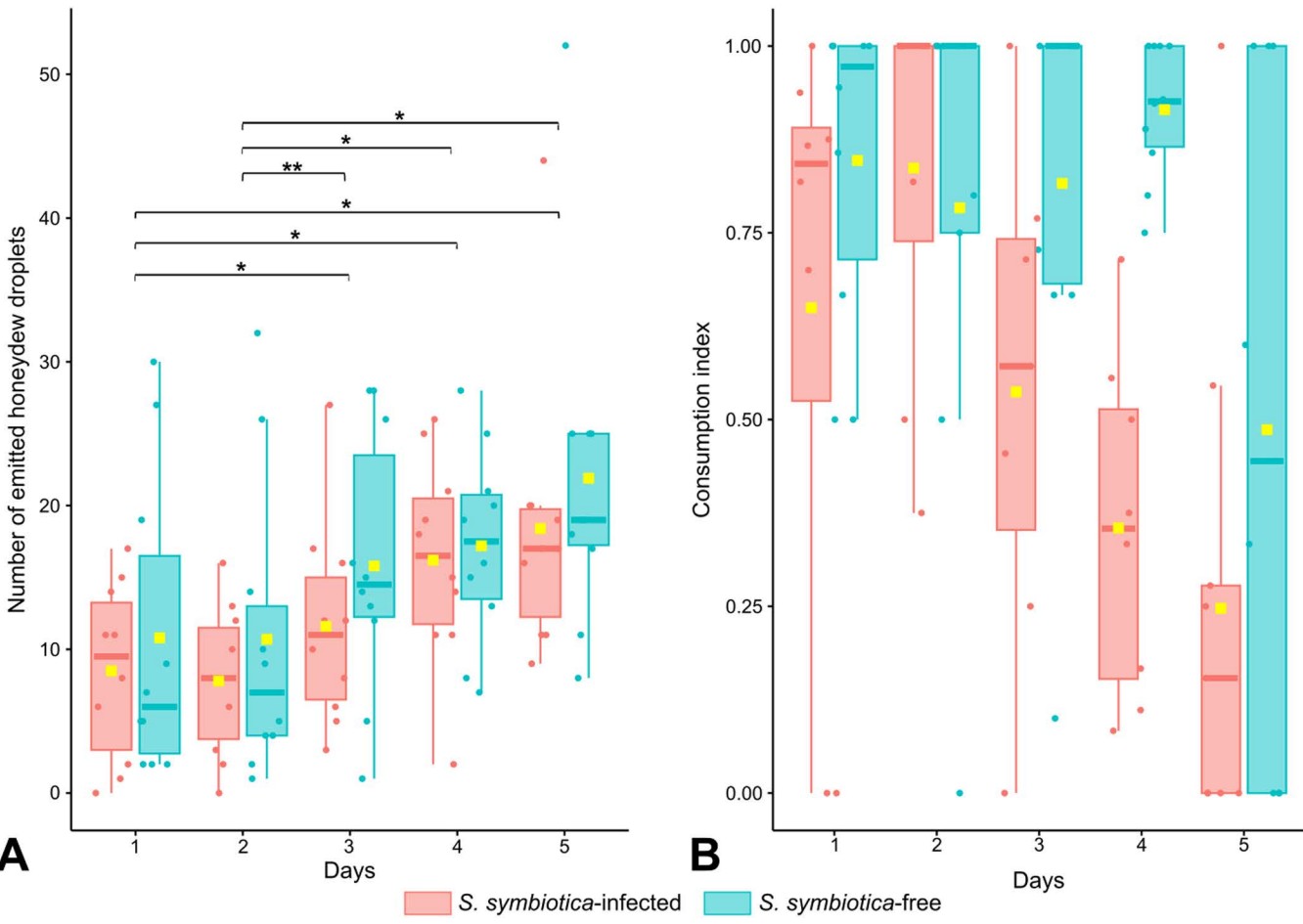

**Fig 3. Honeydew production by aphids and honeydew consumption by ants.** (A) Honeydew production by *S. symbiotica*-free aphids (blue) and *S. symbiotica*-infected aphids (red). Each point represents the total number of droplets released either spontaneously or following ant's stimulation, during 45 minutes of daily observation; N = 10 replicates. Experimental days were compared using Tukey's post-hoc tests and were connected with brackets when being significantly different (*,p < 0.05;**,p < 0.01). (B) Proportion of honeydew droplets ingested by ants emitted by *S. symbiotica*-free aphids (blue) and *S. symbiotica*-infected aphids (red). Each point represents the proportion of honeydew droplets consumed out of all those emitted after stimulation by an ant during 45 minutes of daily observation; N = 10 replicates. The horizontal bar within the boxes represents the median, the upper and lower boundaries of the boxes represent, respectively, the 75th and 25th percentiles. The yellow square is the mean daily production of honeydew (3A), and the mean daily percentage of honeydew droplets consumed (3B),averaged over the 10 replicates.

of droplets consumed after stimulation by ants (GLMM: interaction effect: Wald $\chi^2$ test = 10.7, df = 4, p = 0.03; S7 Table). The ingestion of stimulated honeydew droplets by ants decreased quickly over time for *S. symbiotica*-infected aphids while it remained fairly constant for uninfected aphids for the 4 firsts days (Fig 3B). For both *S. symbiotica*-infected and uninfected aphids, this decrease was the most pronounced on day 5, when the average daily consumption of stimulated droplets decreased by more than half compared to day 1. Thus, ants progressively became less eager to ingest the honeydew droplets they had just stimulated, in particular when being released by *S. symbiotica*-infected aphids. Moreover, the time elapsed between the release of a droplet by an aphid stimulated by an ant and its ingestion was longer for *S. symbiotica*-infected aphids (Log-rank: $\chi^2$ = 19.4, df = 1, p < 0.001; Fig 4). The probability per unit time that an ant would ingest honeydew droplets from uninfected aphids was 1.33 times higher than that from *S. symbiotica*-infected aphids (Fig 4).

**Table 1. Honeydew emission per aphid during the first five days of the experiment.** The daily number of droplets emitted per aphid is given according to its infection status, making a distinction between droplets of which the release was either spontaneous or stimulated by the ants. N values are the numbers of observations made per condition of bacterial infection status and time. Mean and standard error values are provided, as well as p-values of two-ways repeated measures ANOVA (*, p < 0.05; ***, p < 0.001), the ant's colony is treated as within subject factor. For each type of emitted droplets, post-hoc multiple pairwise comparisons (PWC) were performed when effects were significant, and adjusted using Tukey's method, p-values are indicated (*, p < 0.05) and experimental days sharing a common letter are not significantly different.

| | Total number of emitted droplets | | | Number of spontaneously emitted droplets | | | Number of stimulated droplets | |
|---|---|---|---|---|---|---|---|---|
| | *S.symbiotica*-free | *S.symbiotica*-infected | | *S.symbiotica*-free | *S.symbiotica*-infected | | *S.symbiotica*-free | *S.symbiotica*-infected |
| Day 1 | 2.2 ± 1.8 N = 10 | 1.5 ± 0.9 N = 10 | a | 0.9 ± 1.4 N = 10 | 0.3 ± 0.5 N = 10 | ab | 1.3 ± 1.1 N = 10 | 1.2 ± 1.0 N = 10 |
| Day 2 | 2.1 ± 1.5 N = 10 | 1.6 ± 1.0 N = 10 | a | 0.6 ± 0.9 N = 10 | 0.3 ± 0.6 N = 10 | ab | 1.5 ± 1.6 N = 10 | 1.3 ± 1.0 N = 10 |
| Day 3 | 2.9 ± 1.5 N = 10 | 2.5 ± 1.0 N = 10 | ab | 1.3 ± 1.4 N = 10 | 0.9 ± 0.9 N = 10 | b | 1.5 ± 1.3 N = 10 | 1.6 ± 1.5 N = 10 |
| Day 4 | 3.7 ± 1.4 N = 10 | 3.0 ± 0.9 N = 10 | b | 2.1 ± 0.9 N = 10 | 1.3 ± 0.6 N = 10 | bc | 1.7 ± 0.4 N = 10 | 1.8 ± 1.3 N = 10 |
| Day 5 | 3.8 ± 1.5 N = 10 | 3.2 ± 1.4 N = 10 | b | 2.7 ± 15 N = 10 | 1.5 ± 1.0 N = 10 | c | 1.1 ± 1.7 N = 10 | 1.6 ± 1.5 N = 10 |
| Day 1-Day 5 | 2.9 ± 1.7 N = 50 | 2.4 ± 1.3 N = 50 | | 1.5 ± 1.4 N = 50 | 0.9 ± 0.9 N = 50 | | 1.4 ± 1.4 N = 50 | 1.5 ± 1.3 N = 50 |
| | | | | PWC: p = 0.02* | | | | |
| Infection effect | ANOVA II: p = 0.7 | | | ANOVA II: p = 0.02* | | | ANOVA II: p = 0.7 | |
| Time effect | ANOVA II: p < 0.001*** | | | ANOVA II: p < 0.001*** | | | ANOVA II: p = 0.7 | |
| Interaction effect | ANOVA II: p = 0.9 | | | ANOVA II: p = 0.02 | | | ANOVA II: p = 0.8 | |

## Discussion

The multitrophic system involving host plants, aphids and ants has been extensively studied [36–45]. It has revealed the impact of symbiotic bacteria on the fitness of aphids as well as on their interactions with natural enemies [14,19,26–30,35]. However, understanding how these symbiotic bacteria of aphids can influence their ant partners, remains largely elusive. Here, we show that the gut-associated *S. symbiotica* strain alters the growth of aphid populations, regardless of whether they are attended by ants or not. As for its impact on ant mutualists, we found that the presence of *S. symbiotica* bacteria in aphids has no clear deterrent or attractive effects on ants, although ant foraging is slightly reduced on plants hosting *S. symbiotica*-infected aphid colonies. Bacteria also have a direct negative effect on ants' consumption of honeydew, which is known to be the cement of ant interactions with sap-sucking insects. The slower growth of *S. symbiotica*-infected aphid colonies coupled to the lower palatability of their honeydew for the ants suggests that bacteria may modify mutualistic interactions with ants and may ultimately lead ants to forage on other aphid colonies or alternative sugar producers. Social partnerships have a variable influence on a host's microbiome, namely depending on whether microbial transmission is vertical and phylogenetically constrained [75] or horizontal and socially mediated [75,76]. Our findings demonstrate that this influence can also operate in the opposite direction: the host microbiome, particularly the presence of *S. symbiotica* bacteria, can modulate social partnerships by altering trophobiotic relationships between ants and aphids. Specifically, the reduced attractiveness of honeydew produced by *S. symbiotica*-infected aphids aligns with previous observations of a higher prevalence of *Serratia* bacteria in the microbiomes of *Dysaphis* aphids that are not tended by ants (76 but see Henry et al. [77]).

The free-living *S. symbiotica* strain can circulate throughout the whole plant-aphid-ant system [13,60] and may impose fitness costs on each participant of this multitrophic chain. As previously reported, defensive symbionts, i.e., symbionts

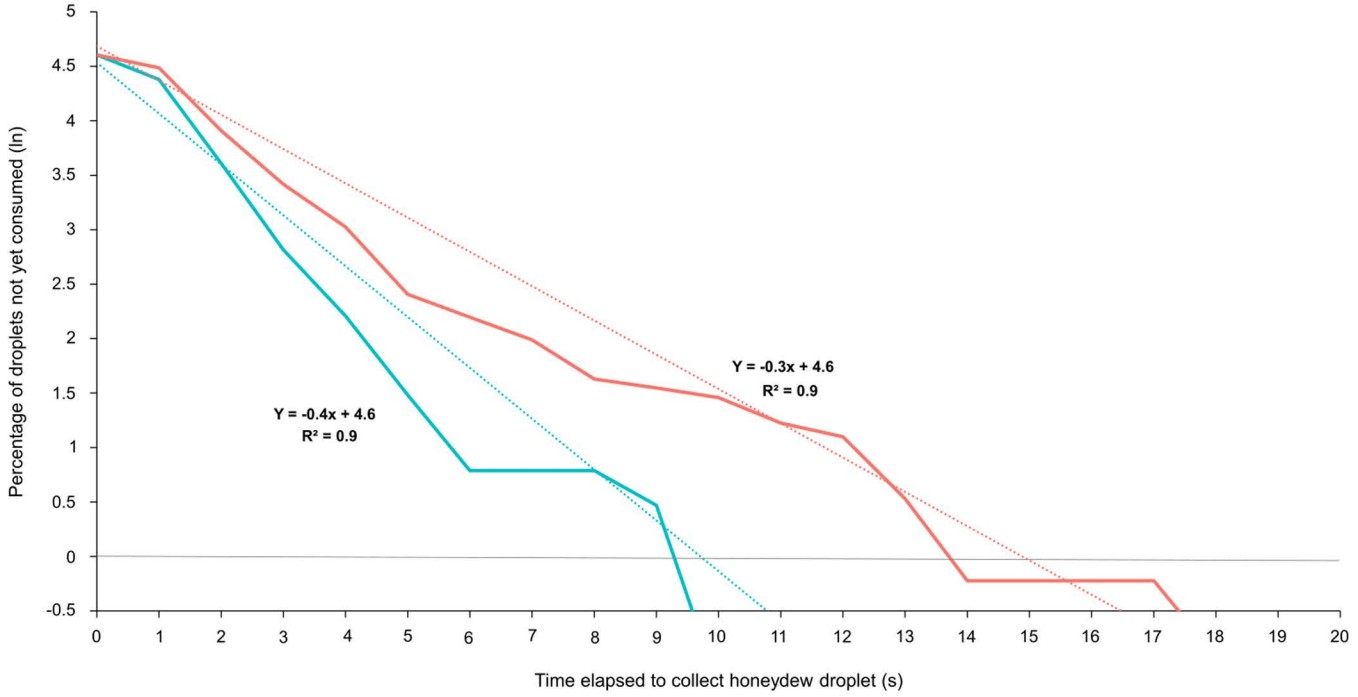

**Fig 4. Honeydew droplets released by aphids as a function of time until consumption by ants.** Survival curves are plotted for the percentage (Ln) of droplets not yet consumed by ants at a given time elapsed since their release by aphids. Honeydew emitted by uninfected aphids (blue, n = 317 droplets consumed) and by *S. symbiotica*-infected aphids (red, n = 232 droplets consumed). The dashed lines represent linear trendlines for each curve. The slope represents the probability of a stimulated droplet to be ingested by ants per time unit.

that enhance aphid protection against parasitoids [26,27,77–80], are associated with costs for their hosts [14,26,77–79]. Indeed, aphids infected by gut-associated *S. symbiotica* strain suffer from reduced survival and fecundity in the absence of stressful conditions [14,79] and exhibit a reduced mass of adults [14]. Although the mechanisms by which *S. symbiotica* bacteria protect their aphid hosts are not well understood [19], we can speculate that the poorer condition of bacteria-infected aphids makes them less attractive to predators and affects the optimal development of parasitoid larvae [14,80]. Our results corroborate these previous literature data, since *A. fabae* populations infected with the gut-associated *S. symbiotica* strain grew slower and spontaneously released less honeydew than the uninfected aphid colonies. Our results also showed a beneficial impact of ants on aphid populations regardless of their infection status: they grew faster than unattended ones, as previously reported in other studies [43–45]. Ants kept their positive effects on *S. symbiotica*-infected aphid colonies, thereby compensating somewhat for the genuine lower growth of aphid population.

Ant-aphid mutualism is a labile relationship, the initiation of which depends on the probability of encountering partners and its maintenance is linked to the cost-benefit balance compared to alternative, possibly more profitable resources [38–40,81]. Facultative symbionts can confer an array of extended phenotypes to their aphid host [14,20–23,26,78,79]. Such changes induced by bacterial symbionts can influence how aphids interact with higher trophic levels. For instance, bacteria can contribute to the release of volatile organic compounds (VOCs) that act as an attractive signal to predators [35]. Likewise, *Staphylococcus xylosus* bacteria make *A. fabae* aphids easier to detect and more attractive to *L. niger* scouts [34]. In this study, we found no significant impact of bacteria on the first steps of foraging as starved scouts, which were highly motivated to explore the nest surroundings, quickly discovered both aphid colonies irrespective of their infection status.

Once aphid colonies have been discovered by ants, a milestone in the maintenance of ant-aphid mutualism is the quality and the amount of honeydew released [37]. The release of honeydew droplets by aphids can be either spontaneous or triggered by prolonged antennal contact by ant foragers. The spontaneous emission of honeydew increased over time in both aphid populations and was significantly more pronounced for the uninfected ones. Given that infection by *S. symbiotica* is known to alter the physiological capacities of aphids [79], future studies should assess whether bacterial infection also influences the movement patterns (based on personal observations, infected aphids tend to move around more frequently), probing behavior and sap feeding behavior of aphids, thereby explaining their lower propensity to spontaneously release honeydew. While the gut-associated bacterial strain appeared to affect the ability of aphids to release honeydew spontaneously, it did not affect the production of honeydew droplets when their emission had been triggered by the ants. This suggests that, if bacteria affect the level of interactions with ants, this was not due to a lower responsiveness of aphids to the stimulation of foragers triggering honeydew production. As for the ants, we found that they were significantly less eager to consume honeydew released by *S. symbiotica*-infected aphids and took more time to harvest droplets. The most plausible explanation for ants' disinterest in consuming honeydew from *S. symbiotica*-infected aphids is that bacteria have altered its composition. After 48 hours of contact with aphids, consumption of honeydew from infected aphids by ants decreased until the fifth day of observation and became, on average, twice lower than for uninfected hemipterans. At this time, corresponding to the 10th day post-infection, *S. symbiotica* bacteria have completely colonized the entire digestive tract of aphids and are probably present in huge quantities both in the aphid gut and honeydew droplets [14,58,59]. Knowing that the sugar content of honeydew strongly influences ant behavior towards aphids [37,82,83], further studies should investigate how *S. symbiotica* bacteria modify the composition of honeydew, as it has already been reported for other endosymbionts [84,85]. In addition, unlike some aphids in which ant attendance increases honeydew quality and quantity [82,86,87], aphids infected with *S. symbiotica* bacteria may be unable to modulate their excretion behavior or honeydew quality due to their altered physiological condition, ultimately reducing their attractiveness to ants as trophobionts. A last non-exclusive explanation is that honeydew droplets become progressively loaded with toxins being metabolites produced by bacteria [88], which can be repellent for the ants. Colonization of *A. fabae* digestive tract by the bacterium could also have modified the viscosity of honeydew droplets, making them less palatable to tending ants.

Despite the decreasing interest in honeydew of *S. symbiotica*-infected aphids, our study did not reveal a disruption in their partnership with ants, which continued to visit both plants for several days, until the end of our experiment. However, we speculate that the reduced honeydew consumption by ants observed at a local level could have negative consequences at the higher level of *S. symbiotica*-infected aphid populations, by reducing the ant attendance and the associated benefits to aphid fitness. Ants, like *L. niger*, are known to adjust the intensity of food recruitment as well as the fidelity to a food source according to its amount, its composition and its persistence [37,89–92]. Especially in mass-recruiting ants such as *L. niger*, the laying of a recruitment trail and its associated snowballing effects allow to focus the activity of the entire ant colony on the most rewarding sources [93,94]. In this study, we did not observe such a selective mobilization of foragers towards uninfected aphids when they got access to both aphid-hosting plants, probably because on the first day of the experiment, the presence of bacteria had not yet altered honeydew consumption. Later on, along with a progressive decline in feeding on *S. symbiotica*-infected honeydew, ants showed lower exploitation and interaction indices on bacteria-infected aphid colonies. Over a longer timescale and under more natural conditions, one may speculate that ants would abandon *S. symbiotica*-infected aphid colonies or at least would redirect a part of their foraging activity to alternative honeydew-producing colonies. Indeed, aphid colonies are usually abundant and do not represent a limiting resource for the ants, which can easily turn their tending behaviors to the most profitable aphid populations [48,90,94]. The impact of *S. symbiotica* on aphid populations is thus complex, with direct costs on their fitness as well as indirect ecological benefits and costs on their natural enemies and ant mutualists respectively. This suggests that the association with protective symbionts may be an alternative way used by aphids to reduce selective pressure from parasitoids when they are not tended and protected by ants. Some studies have already shown an influence of ant attendance on the abundance

of defensive symbionts, such as *Hamiltonella defensa*, *Regiella insecticola* and *S. symbiotica* [75–77,95], those bacteria being more likely to occur in aphid species that are not tended by ants [75–77,95]. Conversely, aphids may limit some costs related to hosting defensive symbionts by attracting ants that deter their natural enemies.

Finally, by altering the feeding behavior of ants, *S. symbiotica* may have unexpected effects on the first level of this multitrophic chain, which is the host plant. Indeed, ants can redirect their foraging activity towards other sources of carbohydrates such as extrafloral nectaries. The broad bean plant, *V. faba*, which was used in our experiments, has extrafloral nectaries (EFNs) at the base of its leaves upon which ant foragers feed. Thus, when searching for carbohydrates, ants have the choice between aphid honeydew and extrafloral nectar, the latter being characterized by a noticeably high sugar concentration [96,97]. Since EFNs produce a quantity of nectar substantially lower than aphid honeydew [98], the number of ants foraging on the plant is likely driven primarily by honeydew availability. EFNs may nevertheless have a significant effect on the spatial distribution of ants across the plant. In cases where *V. faba* is infested by *S. symbiotica*-infected aphids that produce a less palatable honeydew, it is likely that ants will redirect some of their activity towards the plant's EFNs rather than sap-feeding insects. When considering the host plant, infection by *S. symbiotica* thus appears to have several positive effects. First, bacteria increase the development of root network [60]. Secondly, infection by *S. symbiotica* reduces the population growth of sap-sucking aphids, thereby limiting their deleterious effects on the plant. Finally, a redirection of ant foraging towards EFNs due to a lower attractiveness of *S. symbiotica*-infected honeydew would disperse ants over the whole plant, thereby enhancing the plant protection from herbivores [97–100].

In the case of mutualistic endosymbionts, vertical transmission from mother to offspring is a key process for the establishment and maintenance of stable, intimate host-symbiont associations [101]. Conversely, gut bacteria lack such a reliable pathway of maternal transfer [102,103] and *S. symbiotica* strains which retain free-living capabilities, are transmitted via contamination with honeydew or acquired directly from the infected sap of their host plant [14,60]. These horizontal transfers mean that certain *S. symbiotica* strains can circulate between both phylogenetically close and distant species, and can be found in insects associated with aphid colonies, such as parasitoids, predators [50], as well as in the gut of ants tending aphids [13]. Horizontal transfer and social transmission between workers within ant colonies, likely explain why ants tending the same trophobionts often harbor similar bacterial strain in their microbiome [75]. The evolution towards more intimate symbiont-host associations depends on the prevalence and stability of these symbionts in each compartment of the trophic chain as well as on the derived selective advantages conferred to the host. Currently, our understanding of the selective forces involved is partial and highly variable depending on the species in the multitrophic system. For the aphid-tending ants, to our knowledge, we ignore whether a *S. symbiotica* infection of their digestive tract imposes a cost or provides a benefit to ant foragers. For the host plant, existing data converge on increased fitness of *S. symbiotica*-infected plants through enhanced root development [60], lower growth of aphid population, and possibly reduced herbivory due to redirection of ant foraging towards extrafloral nectaries. For the aphids, gut colonization by *S. symbiotica* bacteria was hypothesized as a possible evolutionary step toward the stabilization of long-term association with the hemipterans. However, the cost-benefit balance resulting from *S. symbiotica* infection is questionable for the aphids and probably context dependent. There are many cases of endosymbionts whose presence is costly to their host but that become mutualistic partners when exposed to specific selection pressures [18–22,49]. Similarly, while *S. symbiotica* infection has negative effects on several *A. fabae* life history traits in the absence of natural enemies, it is plausible that these gut bacteria protect aphids from other environmental stressors or play a nutritional role given the conservation of a large repertoire of genes related to metabolism [55,79,104]. This study has revealed a new cost of bacterial infection for the aphids, which is the progressive disinterest of ants for their honeydew, possibly leading to a disruption of ant-aphid partnership and to the loss of associated benefits, such as reduced aphid hygiene and weaker defense against natural enemies. Based on these results, we speculate that the ants will not promote the occurrence and dispersal of *S. symbiotica* bacteria, as reported in natural populations of aphids in which defensive symbionts are not fixed and occur at low to intermediate frequencies [26,28,77].

In conclusion, this study contributes to our understanding of the evolution of bacterial mutualism in aphids, from free-living bacteria acquired from the environment to intracellular endosymbionts. The weakening of ant-aphid interactions suggests that bacterial infection may tip the balance towards higher costs than benefits for the aphid host. Rather than representing an evolutionary step towards a closer endosymbiotic association, aphid gut-associated *S. symbiotica* may simply constitute a reservoir from which beneficial associations with other participants in the multitrophic systems can emerge. As suggested by Pons et al. [60], sap-sucking aphids could be vectors for horizontal transfer of bacteria that confer beneficial biological effects mainly to the host plants. These plants appear to be a permanent source of contamination, which is required to maintain infection across aphid generations and to establish new symbiotic associations across multiple insect species, including caterpillars that feed on the same host plant as aphids. Elucidating the factors and players that influence horizontal transfer and host jumping of certain bacterial strains remains a major challenge within the field of symbiosis research.

## Supporting information

**S1 Fig. Schematic representation of the experimental set-up.**
(TIF)

**S2 Fig. Daily ascending flows of ants towards the plant hosting either *S. symbiotica*-free aphids (blue) or *S. symbiotica*-infected aphids (red).** Average values are represented by points and were calculated over 10 replicates for each infection status, error bars represent standard deviation.
(TIF)

**S3 Fig. Duration of antennal contacts performed by ants to *S. symbiotica*-free aphids (left panel) and to *S. symbiotica*-infected aphids (right panel), observed over the 5 first days of experiment.** Each grey point represents a contact event, n = 364 on *S. symbiotica*-free aphids and n = 380 on *S. symbiotica*-infected aphids over 10 replicates for each infection status. The number of antennal contacts observed are indicated for each day. The blue trend line is modeled by "*flexplot*" function in R with the assumption of a Gaussian distribution and shows the general pattern of duration of antennal contact over time.
(TIF)

**S1 Table. Description of models used for statistical analyses.**
(DOCX)

**S2 Table. Model comparison for the growth dynamics of aphid populations, considering the infection status, time and presence of ants as fixed factors as well as first and second order interaction effects.** Models were compared using maximum likelihood estimate of the model (log10L), Akaike's Information Criterion corrected (AICc). Delta AIC and the degree of freedom (df) are also indicated. Models are ranked by increasing values of AICc. The arrow indicates the best model considered.
(DOCX)

**S3 Table. Model comparison for ascending flow of ants, with the infection status and time as fixed factors as well as first order interaction effect, the random factor for all models is the ant colonies.** Models were compared using maximum likelihood estimate of the model (log10L), Akaike's Information Criterion corrected (AICc). Delta AIC and the degree of freedom (df) are also indicated. Models are ranked by increasing values of AICc. The arrow indicates the best model considered.
(DOCX)

**S4 Table. Model comparison for the proportion of consumed droplets by ants, with the infection status and time as fixed factors as well as first order interaction effect, the random factor for all models is the ant colony.** Models

were compared using maximum likelihood estimate of the model (log10L), Akaike's Information Criterion corrected (AICc). Delta AIC and the degree of freedom (df) are also indicated. Models are ranked by increasing values of AICc. The arrow indicates the best model considered.
(DOCX)

**S5 Table. Model comparison for the duration of antennal contacts, with the infection status and time as fixed factors as well as first order interaction effect, the random factor for all models is the ant colony.** Models were compared using maximum likelihood estimate of the model (log10L), Akaike's Information Criterion corrected (AICc). Delta AIC and the degree of freedom (df) are also indicated. Models are ranked by increasing values of AICc. The arrow indicates the best model considered.
(DOCX)

**S6 Table. P-values of Wilcoxon signed rank tests with Bonferroni adjustment, comparing the effects of aphid infection status per day and the effects of attendance by ants per day (\*, p < 0.05; \*\*, p < 0.01; \*\*\*, p < 0.001).** As for the time effect, experimental days that shared a common letter were not significantly different when using Tukey's post-hoc tests.
(DOCX)

**S7 Table. Consumption of honeydew droplets by ants over the course of the experiment.** N values are the numbers of observations per infection status of aphids and time for the proportion of droplets consumed by ants after having stimulated honeydew emission and for the duration of antennal contacts. Means and standard error values are provided,as well as p-values for fixed factors and interaction effect (\*, p < 0.05; \*\*,p < 0.01; \*\*\*, p < 0.001), the ant's colony is used as random factor. Post-hoc multiple pairwise comparisons (PWC) were conducted and adjusted using Tukey's method, p-values are indicated (\*, p < 0.05; \*\*, p < 0.01; \*\*\*, p < 0.001) and experimental days sharing a common letter are not significantly different.
(DOCX)

## Acknowledgments

We would like to thank Luc Dekelver for helping us to collect ants on the field and his technical support and to thank Prune Gallezot for helping us to conduct experiments and collect data. We also thank Linda Dhondt and Laurent Grumiau for helping us to carry out PCR analyses.

## Author contributions

**Conceptualization:** Margaux Jossart, Thierry Hance, Claire Detrain.

**Data curation:** Margaux Jossart.

**Formal analysis:** Margaux Jossart, Claire Detrain.

**Funding acquisition:** Thierry Hance, Claire Detrain.

**Investigation:** Margaux Jossart.

**Methodology:** Margaux Jossart, Thierry Hance, Claire Detrain.

**Supervision:** Thierry Hance, Claire Detrain.

**Validation:** Margaux Jossart, Claire Detrain.

**Visualization:** Margaux Jossart, Claire Detrain.

**Writing – original draft:** Margaux Jossart, Claire Detrain.

**Writing – review & editing:** Margaux Jossart, Thierry Hance, Claire Detrain.

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
