## [Decision Letter · Decision Letter 0]

Dear Dr. Jossart,

First of all, I apologize for the time this process has taken. Thank you for submitting your manuscript to PLOS ONE. After careful consideration, we feel that it has merit but does not fully meet PLOS ONE’s publication criteria as it currently stands. Your article was reviewed by two experts who confirmed that your study is very interesting and deserving of publication. However, they also pointed out some shortcomings that I invite you to address. Therefore, we kindly invite you to submit a revised version of the manuscript that addresses the points raised during the review process. I would appreciate it if you could send a document with the suggested changes.

We look forward to receiving your revised manuscript.

Kind regards,

Clara F. Rodrigues

Academic Editor

PLOS ONE

Journal Requirements:

Reviewers' comments:

Reviewer's Responses to Questions

**Comments to the Author**

1. Is the manuscript technically sound, and do the data support the conclusions?

Reviewer #1: Yes

Reviewer #2: Yes

2. Has the statistical analysis been performed appropriately and rigorously?

Reviewer #1: Yes

Reviewer #2: Yes

3. Have the authors made all data underlying the findings in their manuscript fully available?

Reviewer #1: Yes

Reviewer #2: Yes

4. Is the manuscript presented in an intelligible fashion and written in standard English?

Reviewer #1: Yes

Reviewer #2: Yes

Reviewer #1: A very interesting manuscript on an experiment, that sheds new light on microbiome-driven factors affecting ant-aphid mutualism. The experiment and the analyses of the results seem to be well-prepared, as well as the manuscript, although the quality of the figures could be much better for publication. I have only a few more suggestions on the manuscript:

Line 46, 49 – add the spaces before the brackets

Line 66 – remove the space before the comma

Line 70 – ant-aphid mutualism also influenced aphid evolution, for details please see: Depa et al. 2020

Line 135-140 – did the ant colonies contain a queen and brood? As there were nursery workers, I assume there were larvae and the demand for nutrients was constant. If not, then the ants perhaps have no occasion to utilize the energy from honeydew, and their demand for food decreased during the study.

Now we also know that ants do eat aphids they tend (e.g. Sakata 1994). Were there any cases of preying on attended aphids, which might have influenced the rate of the aphid colony growth?

Line 148 – remove the space before the dot

With the M&M section, I realise that, as you stated, certain interrelations between ant-ant aphids ceased to occur after Day 5, but providing these further results somehow would highlight the meaning of these relations while they occurred, just by setting the broader background.

Line 506-509 – I have a feeling that it might be safer to use the term “modify” than “weaken”. A. fabae is facultatively myrmecophilous, which means there are cases when they are aggressively defended by ants, and there are cases when they are not ant-attended at all, and that does not mean their colonies are in bad condition. In your case, indeed, Serratia weakened the mutualistic engagement of ants, but it is too weak a result to generalize it to the statement that it may weaken the ant-aphid mutualism. Unless, of course, we prove that many more facultatively myrmecophilous aphid species do have Serratia when unattended and no Serratia (or some other bacteria) when ant-attended.

For the purpose of the Discussion, it might be beneficial to refer to Kaszyca-Taszakowska & Depa 2022 paper, where also aphids unattended by ants have a higher proportion of Serratia in their facultative microbiome. The results of the submitted manuscript may largely explain that report, via different explanations of the influence of Serratia on ant-aphid mutualism. Additionally, I would suggest referring to the paper by Ivens et al. 2018, where Serratia issues may also be interconnected with the data in the manuscript.

Reviewer #2: After numerous studies on the effects of endosymbiotic bacteria on aphid ecology, your research provides valuable insight into the effects of gut-colonizing bacteria on aphid populations and their interactions with aphid-tending ants. I found this well-designed study to be highly interesting, and I am intrigued by the idea that the boundaries between plant-benefitting bacteria and aphid symbionts can be so blurred. However, I have identified several points that need to be addressed prior to publication.

Introduction:

Your introduction is compelling and requires only minor corrections (see comments below). Please double-check your punctuation, as there are numerous comma errors that I did not highlight individually.

Methods:

You have described the procedure for infecting aphids with the bacterial strain in detail, but it is unclear whether you verified the success of this procedure. Typically, a diagnostic PCR is performed to confirm infection status before proceeding with experiments involving endosymbiotic bacteria. Do you have data demonstrating that your infection procedure is sufficiently reliable to justify not testing infection status beforehand?

I also found your aphid counting methods somewhat unclear. Why did you count only adults and third instar nymphs, despite A. fabae having four nymphal stages? This methodology may explain why population growth only began on day 5, when the first nymphs reached late instars. If there is a valid reason for omitting early instar aphids, please mention it and clarify the rationale for this approach.

Results:

Would it be possible to merge certain figures (e.g., Figures 3, 4, and 5) and move relevant supplementary data into the main text? Doing so could improve readability and reduce the need for frequent cross-referencing. Additionally, consider refining the layout of figures to enhance clarity (see detailed comments below).

Use consistent terminology throughout the results section (e.g., always use “infection status”), and present results in a logical order, such as addressing main effects before interaction effects (see specific comments below).

Discussion:

Your discussion is engaging and generally well-written. However, there are a few points that I recommend addressing or expanding upon:

1. Consider discussing the effect of ants on aphid honeydew quality and excretion (see comments on line 560). Previous studies suggest that ant-tending increases both the quality and quantity of honeydew. It is worth considering whether S. symbiotica-infected aphids were unable to improve honeydew quality over time, leading to reduced ant attention.

2. Highlight the potential confounding effect of Serratia infection in both aphids and plants on ant behavior and plant occupation (see comments on lines 602–608). Given that ants might be collecting nectar from EFNs instead of honeydew from aphids, you should either acknowledge this limitation or explain why your plant occupancy index remains valid.

Finally, please review the punctuation in the discussion section, as I noticed several comma errors that I did not mark individually.

Specific comments and suggestions:

Line 34: …large amounts (plural)

Line 51: remove the “a” before better

Line 63: The comma needs to be before the “but” and not after

Line 90: …exhibit variable degrees of… (no “a” and degrees in plural)

Line 103: I would suggest rewriting the middle part of this sentence: …(29, 54-56) , allowing manipulation of its occurrence in…

Lines 135-136: …aphid-tending species that is widespread in temperate regions of Europe.

Line 145: To my knowledge Christoph Vorburger is a Professor at ETH Zürich and not at the University of Zürich.

Line 168: Finally with double l.

Line 171: …protocols found in the literature.

Line 180: …except that sterile PBS … (not excepting)

Lines 207-208: …before the beginning of the experiment… (the missing)

Line 240: The recorded images covered an area… (no “on” needed)

Line 252: , we measured the time elapsed between… (time is not a plural)

Line 281: We included the infection status… (not infectious)

Line 296: infection status (not infected status)

Line 329: Always use same terminology to improve clarity. S. symbiotica-free and S. symbiotica-infected

Lines 337-339: Move this sentence to the methods section.

Lines 377-379: Move this sentence to the methods section.

Lines 416-417: Unclear sentence. Suggestion: Indices significantly lower than the theoretically expected value of 50%.

Lines 421-426: Mention the main effect before the interaction effect. Is “infection status” the same as “infection condition”? Please use consistent terminology throughout the manuscript.

Line 452: …according to the best… (to missing)

Line 453: We however found a significant… ("a" is missing)

Lines 478-479: I think there is a word missing in this sentence. (…aphids would be(?) twice that of…)

Line 560: Maybe mention that the presence of ants can also change the composition of released honeydew (shown in your reference 76). Maybe S. symbiotica infected aphids were not able to increase honeydew quality over time and thus received less ant attention.

Lines 602-608: According to what you say here, ant behaviour can be influenced by the effect of the bacteria on the aphid or on the plant since plants can also get infected with this Serratia strain. This confounding effect impacts your plant occupancy index because ants observed on a plant might be collecting nectar from EFNs rather than honeydew from aphids. Please mention this restriction and/or clarify why your plant occupancy index is still useful.

Table S1: replace “Colonial ID” with “Colony ID”

Figures: If there is space, I would find it clearer to add a legend to the plots than describing the meaning of the colours in each caption.

Figure 4: You mention in the caption that “Significant differences are shown for pairwise comparisons…” but I cannot see any bars or lines indicating differences in the Figure.

Figure 5: Why does this figure have horizontal lines while none of the others does? Please adapt.

**Do you want your identity to be public for this peer review?** For information about this choice, including consent withdrawal, please see our Privacy Policy

Reviewer #1: No

Reviewer #2: No

---

## [Author Response · Author response to Decision Letter 1]

8 May 2025

We would like to thank the editor as well as the two reviewers for their constructive comments and insightful suggestions. We have addressed all comments that were raised and responses are given in the rebuttal letter (see Responses to reviewers.docx)

---

## [Editor Report · Decision Letter 1]

Infected Connections: Unraveling the Impact of a Bacterial Symbiont on Ant-Aphid Partnership

PONE-D-24-46528R1

Dear Dr. Jossart,

We’re pleased to inform you that your manuscript has been judged scientifically suitable for publication and will be formally accepted for publication once it meets all outstanding technical requirements.

Kind regards,

Clara F. Rodrigues

Academic Editor

PLOS ONE
---

## [Editor Report · Acceptance letter]

PONE-D-24-46528R1

PLOS ONE

Dear Dr. Jossart,

I'm pleased to inform you that your manuscript has been deemed suitable for publication in PLOS ONE. Congratulations! Your manuscript is now being handed over to our production team.

Kind regards,

on behalf of

Dr. Clara F. Rodrigues

Academic Editor

PLOS ONE